# MEMORISABLE PROMPTING: PREVENTING LLMS FORGETTING FALSE POSITIVE ALARM

## ABSTRACT

Large Language Models (LLMs) are widely recognized for their superior performance across various domains. However, their tendency to generate inaccurate or misleading responses presents significant challenges, particularly in the natural language domain. This issue underscores the need to enhance both the explainability and reliability of LLMs. While recent advancements in prompting have focused on leveraging in-context learning—such as providing step-by-step explanations—these approaches often overlook the critical importance of understanding the response dependency of LLMs on specific datasets. This understanding is crucial for interpreting their outputs and improving their consistency. Moreover, if we can capture and encode these response dependencies, we can integrate them into LLMs as memorized knowledge to mitigate false positive responses over time. In this paper, we tackle this challenge by introducing the Memorizable Prompting (MP) paradigm, which enables LLMs to retain and utilize information from past responses. Specifically, our approach leverages hint samples—a small set of annotated examples—to learn the response dependencies, defined as the relationship between LLM outputs and the ground-truth annotations for a given dataset. This equips LLMs with the ability to recall past false positives and use that knowledge for self-correction in future predictions. We have evaluated our method on a diverse set of domain-specific datasets, demonstrating its effectiveness across large-scale benchmarks.

## 1 INTRODUCTION

Large Language Models (LLMs), such as ChatGPT, have recently gained immense popularity due to their remarkable capabilities across various domains. However, since LLMs (Devlin et al., 2018; Touvron et al., 2023; OpenAI, 2023) are trained on open-domain knowledge, the generated outputs often exhibit randomness, leading to unreliable or inconsistent predictions. These issues are particularly problematic in high-stakes domains that require high-quality supervision, such as recommendation systems and text mining. Therefore, it is vital to devise effective approaches to address these challenges and enable LLMs to generate more consistent and accurate predictions. One potential solution is the use of prompting strategies. Research studies (Brown et al., 2020; Wei et al., 2021; Yao et al., 2022; Liu et al., 2023) have shown that the effectiveness of LLMs is intricately tied to the prompting strategy employed for each specific task. However, these methods can only be considered "static prompting," meaning they do not enable LLMs to self-reflect and self-correct their responses. Consequently, many studies have focused on enabling LLMs to perform self-correction (Bang et al., 2023; Huang et al., 2023; Li et al., 2023; Wei et al., 2024; Zhou et al., 2023; 2022). The self-correction simply means using LLMs to do the self-verification and self-correction. Despite these efforts, Huang et al. (2023) has shown that intrinsic self-correction is not yet achievable using only the self-generated output. Recently, Chen & Tsang (2024) proposed utilising hint samples, a method that allows LLMs to iteratively learn from their consistent samples, thereby improving their predictions. This approach uses hint samples as relevant demonstrations (Shi et al., 2023), but it lacks the memorisation capability necessary for effective

reflection. Memorisation can be understood from various perspectives; in our problem setting, it refers to the ability to memorise a response dependence. It can be defined as the relationship between an LLM's predictions and the ground truth annotations for a given query, such as $X$. Response dependence includes false positive predictions and true positive prediction. It can be used to refine the selection criteria to improve predictions in subsequent rounds. According to (Attias et al., 2024) memorization plays an essential role for good generalization. However, in the prompt based task, the memorisation related prompt based task of LLMs remains limited. If LLMs could memorise past mistakes, they would likely not make the same false positive predictions again. In this paper, We demonstrate the important part of memorisation in prompt based task. This leads us to consider:

*"How can we design a prompting scheme that enables Large Language Models (LLMs) to remember false positive predictions as alarm, thereby improving response accuracy and consistency?"*

To achieve this, we propose the **Memorisable Prompting (MP)** method. MP aims to accomplish two primary objectives. Firstly, it provides a paradigm that allows LLMs to understand their response dependence. Secondly, leveraging these patterns by storing them in a memory bank, thereby endowing LLMs with memorisation capabilities. The goal is to use the memory bank to prevent LLMs from repeating false positive predictions, generating more consistent outputs from the LLMs. Specifically, MP uses small amounts of annotated samples as hints to obtain the label dependence between LLM predictions and ground truth for each query. This dependence is stored in a memorisation bank and used to prevent LLMs from forgetting false positive predictions, enhancing their ability to self-reflect and self-correct, subsequently improving prediction accuracy. In the subsequent stage, for every query and prediction generated using ChatGPT, we will retrieve potential correct label candidates from the memory bank based on the prediction for the large unsupervised training samples. For a more detailed explanation, please refer to *Section 3*. Our experimental results validate the effectiveness of the Memorisable Prompting method. The main contributions of this paper have been listed below:

- Introduction of Memorisable Prompting Method: We introduce the Memorisable Prompting method, which endows LLMs memorising their past false positive prediction. This approach facilitates LLMs to circumvent repetitive false positive predictions by recording and encoding these response dependence into the Memorisable mask matrix.

- Improvement of adaptability and Reliability: We have justified our approach through the lens of probability to interpret the importance of the Memorisable masking matrix. We have provided a new perspective on exploiting the response dependence of LLMs to improve their explainability and reliability, making them more robust and adaptable across various tasks.

- Demonstrated Effectiveness Across Large-Scale Datasets: We have verified the Memorisable Prompting method on various domain datasets. Our experiments show its effectiveness in improving the performance of LLMs across large-scale datasets with a large number of classes, showcasing its potential for broad applicability.

## 2 RELATED WORKS

Prompt-based learning was initially studied in (Brown et al., 2020). It uses few-shot examples as illustrations to aid large language models (LLMs) in generating more refined predictions. Subsequently, (Wei et al., 2021) proposed instruction fine-tuning to improve the performance of LLMs. The ReAct method (Yao et al., 2022) combines chain-of-thought prompting with action. It endows generating more coherent supplemental data from external information, such as the Wikipedia API, to circumvent problems such as hallucinations resulting from chain-of-thought reasoning. Active Prompt (Diao et al., 2023) proposes a question selection approach named 'active-prompt' to improve the accuracy of generated predictions. The 'Generate Knowledge Prompting' method (Liu et al., 2023) uses external information to enable the model to generate more accurate

responses. Consistency Prompting (Wang et al., 2023) aims to improve the response accuracy of LLMs by considering consistently generated answers through selecting multiple and diverse paths in a few-shot chain of thought approach. The problem with this method is its dependence on multiple sources of paths; even slight changes in one source's prediction can drastically impact the final prediction. Chain of Thought Prompting: The chain of thought method (Wei et al., 2022) provides step-by-step illustrations for the given query to the LLMs.Few-Shot Thought Prompting: (Brown et al., 2020) uses a few relevant examples as illustrations in the prompt to aid the model in self-correction. Tree of Thought Prompting (Long et al., 2023; Yao et al., 2023) (Yao et al., 2024; Long, 2023) proposes the Tree of Thoughts (ToT) prompt, which encourages Large Language Models (LLMs) to engage in step-by-step exploration and self-correction by simulating different agents that communicate and provide critiques for each other. Our work is also related to Self-Reflection Prompt-Based Learning. (Huang et al., 2022)suggests a self-correction method that requires LLMs to question their initial responses before providing a final answer. Similarly, (Madaan et al., 2024) proposes generating feedback for its output and using it iteratively to refine itself. However, recent work (Huang et al., 2023) has shown that intrinsic self-correction is not yet achievable. Additionally, since the generated feedback is evaluated solely by the LLMs themselves without any external validation, the trustworthiness of this feedback can be questionable. Unlike (Shinn et al., 2024), which proposes an iterative feedback refinement strategy, our approach involves storing feedback in short-term and long-term memory.

## 3 MEMORISABLE PROMPTING PROBLEM SETTING

### 3.1 PRELIMINARIES

Consider a feature space $\mathcal{X} \subseteq \mathbb{R}^d$ and a label space $\mathcal{Y} = \{1, \ldots, c\}$, where $c$ denotes the number of classes. Each instance $X \in \mathcal{X}$ has a true label $Y \in \mathcal{Y}$. In many real-world scenarios, full supervision is unavailable for the entire dataset. Instead, there is usually a small subset of cleanly labelled samples alongside a larger set of unlabeled data. More specifically, we can define $D_{\text{small}}$ as the distribution of the cleanly labelled small sample set, denoted as hints, which contains pairs $(X, Y, \vec{Y})$ where $\vec{Y} = \mathcal{Y}$, representing a candidate label set encompassing all class labels. This can be expressed as $\{(X_i, Y_i, \vec{Y})\}_{i=1}^s$, with $s$ being the total number of clean samples. We define $D_{\text{large}}$ as the distribution for the large unsupervised dataset, denoted as $\{X_i, \vec{Y}\}_{i=s+1}^n$, where $n$ is the total number of both labelled and unlabeled training samples. The $D_{\text{large}}$ is considered an unsupervised dataset, meaning neither no ground truth label nor weak supervision is associated with each instance in the distribution. The learning objective is to design a prompting strategy that leverages $D_{\text{small}}$, which constitutes about $5\%$ of the total training samples, to allow large language models (LLMs) to accurately annotate the large unsupervised dataset $D_{\text{large}}$.

### 3.2 ENDOWING LLMS WITH MEMORISATION USING ESTIMATED MEMORISATION MASKING MATRIX

In this section, we introduce a hints-based Memorisation masking matrix to mimic the memorisation bank for LLMs, aiming at preventing LLMs from forgetting the false positive predictions in the API-based prompting task. Specifically, we denote a memorisation bank as $M(x)$, a transition matrix that captures the dependencies between a specific true class $Y$ and predicted classes $Y' \in \mathcal{Y}$ for instance $x$. Given $x$ and $Y$, we record all possible $Y'$ from ChatGPT. This setup enables the construction of $M(x)$, representing the dependency between $Y$ and $Y'$ given $x$. We record the frequency of predictions from ChatGPT for each $Y$, noting each occurrence as a potential candidate label. Ideally, the matrix's diagonal entries will be positive (indicating that $Y$ and all corresponding $Y'$ from ChatGPT are the same), while other entries remain zero, signifying correct classification by ChatGPT. However, mismatches often occur between the true label and ChatGPT's predictions, reflected by non-zero off-diagonal entries. Each column of $M(x)$ represents the dependency of the truncated candidate label set on $Y$. For illustration purposes, consider the following example of the

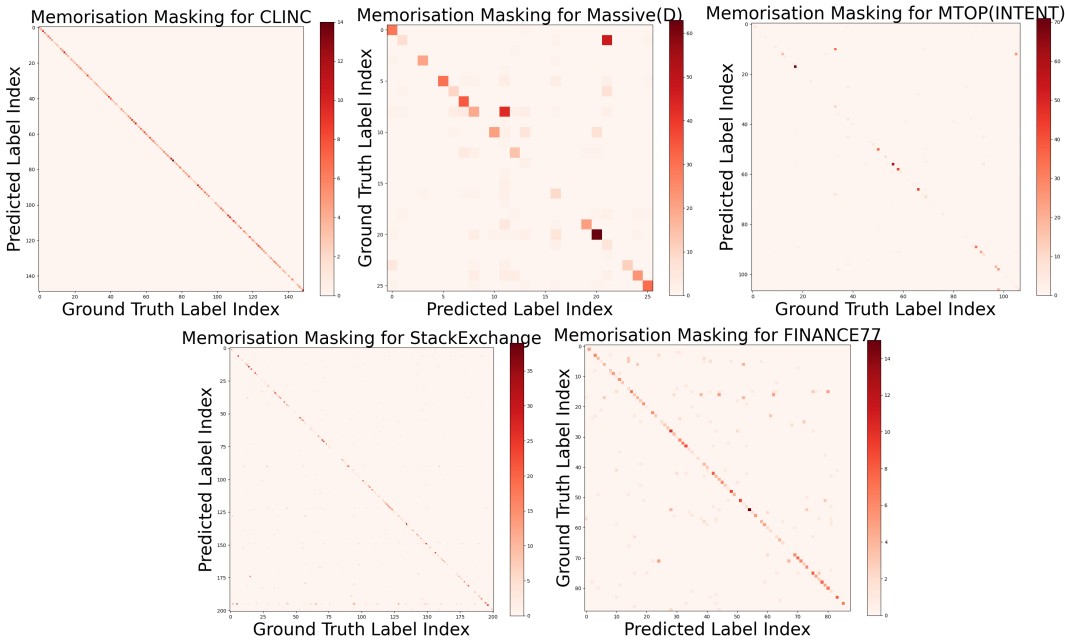

Figure 1: Estimated Memorisation Masking for datasets CLINC, MTOP (INTENT), Massive (Domain), MTOP (INTENT), Stack-Exchange, and Finance77. The diagonal line indicates the correct prediction from LLMs given each dataset, whereas the other highlighted entries indicate false positive predictions.

masking matrix $M$:

$$M(x) = \begin{bmatrix} m_{Y_1'Y_1} & m_{Y_1'Y_2} & m_{Y_1'Y_3} & m_{Y_1'Y_4} \\ m_{Y_2'Y_1} & m_{Y_2'Y_2} & m_{Y_2'Y_3} & m_{Y_2'Y_4} \\ m_{Y_3'Y_1} & m_{Y_3'Y_2} & m_{Y_3'Y_3} & m_{Y_3'Y_4} \\ m_{Y_4'Y_1} & m_{Y_4'Y_2} & m_{Y_4'Y_3} & m_{Y_4'Y_4} \end{bmatrix} = \begin{bmatrix} 1 & 1 & 0 & 0 \\ 0 & 1 & 1 & 0 \\ 0 & 0 & 1 & 1 \\ 1 & 0 & 0 & 1 \end{bmatrix} = \begin{bmatrix} --M_{Y_1'}-- \\ --M_{Y_2'}-- \\ --M_{Y_3'}-- \\ --M_{Y_4'}-- \end{bmatrix}$$

In this case, $m_{Y_1'Y_1} = 1$ and $m_{Y_4'Y_1} = 1$ indicate that given the ground truth label is $Y_1$, the prediction classes are $Y_1'$ and $Y_4'$. On the other hand, $m_{Y_2'Y_1} = 0$ and $m_{Y_3'Y_1} = 0$ suggest that ChatGPT did not predict $Y_2'$ or $Y_3'$ given $Y_1$. $M(x)_{Y',Y}$ indicates whether $Y'$ is a candidate label for $Y$. Each row of the matrix $M(x)$ corresponds to a candidate label $Y'$, and each column corresponds to a true label $Y$. The value at $M(x)_{Y',Y}$ is 1 if $Y'$ is a candidate for $Y$, and 0 otherwise. Overall, by estimating a discrete memorization masking matrix $M$ from hint samples, we construct deterministic mappings between the LLM's generation $Y'$ and the ground truth labels $Y$. Given new inputs, we exploit $M$ to infer potential ground truth labels based on the LLM's generations, refining our candidate sets and improve the LLM's accuracy for subsequent generation.

## 4 PREVENTING LLMs FORGETTING WITH ESTIMATED MEMORISATION MASKING THROUGH THE LENS OF PROBABILITY

Let $G$ denote the LLM, which generates a prediction $Y'$ given a query $X$ and a set of all label candidates $\vec{Y}$, the $G$ selects a label from the $\vec{Y}$ given $X$. The process can be formulated as:

$$P(Y'|X, \vec{Y}) = G(X, \vec{Y}), \tag{1}$$

---

**Algorithm 1** Memorisable Prompting

1: **Input:** Small clean samples $D_{\text{small}} = \{(X_i, Y_i)\}_{i=1}^s$, unlabeled data $D_{\text{large}} = \{X_i\}_{i=1}^N$, total number of categories $K$.
2: **Output:** Refined predictions $\bar{Y}_{\text{Refined}}$
3: **Stage 1: Initialization**
4: Generate initial candidate set $\vec{Y}_{\text{Query}}$ initialized with all ones
5: **Stage 2: Construct Memorisation Masking**
6: Estimate initial mask from noisy similar data pairs
7: Update $\vec{Y}_{\text{Query}}$ using the estimated mask
8: **Stage 3: Predict Ground Truth Labels**
9: Use $D_{\text{small}}$ to obtain initial predictions
10: **Stage 4: Estimate Memorisation Masking**
11: Estimate memorisation mask based on the occurrence of potential correct candidate set labels and predicted labels.
12: **Stage 5: Update Candidate Set**
13: **for** $i = 1$ to $s$ **do**
14:   Update $\vec{Y}_i$ based on the corresponding row in the memorisation mask
15: **end for**
16: **Stage 6: Refining Predictions**
17: Refine predictions $\bar{Y}_{\text{Refined}} = G(X, \vec{Y}_{\text{Updated}})$
18: **return** $\bar{Y}_{\text{Refined}}$

---

where $Y'$ is a prediction from the set $\vec{Y}$, and $P(Y'|X, \vec{Y})$ represents the posterior probability. The learning objective aims to optimise the parameter of $G$ through a prompting strategy to obtain $P(Y|X, \vec{Y})$, where $Y$ is the underlying ground truth label. In addition, the output from ChatPGT is only discrete output either in text or one hot label vector depending on the request of the query. According to equation 1, we can marginalise over $Y$ by deriving $P(Y'|X, \vec{Y}) = \sum_Y P(Y'|Y, X, \vec{Y})P(Y|X, \vec{Y})$. We aim to solve $P(Y|X, \vec{Y})$. Given $P(Y'|X, \vec{Y})$ is known, and if the $P(Y|Y', X, \vec{Y})$ can be obtained, our problem can be solved.

$$P(Y'|X, \vec{Y}) = \sum_Y P(Y', Y|X, \vec{Y}) \tag{2}$$

$$\sum_Y P(Y', Y|X, \vec{Y}) = \sum_Y P(Y'|Y, X, \vec{Y})P(Y|X, \vec{Y})$$

In this paper, we assume the universal Memorisable masking matrix for the whole dataset $D_{large}$ and denote it as $P(Y'|Y, X, \vec{Y})$. Because of that, large language models, such as ChatGPT, have evolved to be so powerful that they can even differentiate nuanced differences within a category, not to mention that individual instances manifest unique features. Mathematically, we have formulated the universal Memorisable masking assumption as follows:

$$P(Y'|Y, X, \vec{Y}) = P(Y'|Y, X', \vec{Y}), \tag{3}$$

where $X$ and $X'$ represent different distinct features of the category $Y$. Intuitively, one might expect a specific instance-dependent transition matrix for each instance to represent the label $Y'$'s generation process given instance $X$. However, we argue that this is generally not the case if a category $Y$ within the textual type dataset typically exhibits unique and distinct features, denoted as $X$ and $X'$. Some category features are so unique that they completely differ from those of other categories. Assuming that LLM can detect the subtle difference between two instances, there should be no problem with correctly predicting the input. For instance, considering a category $Y$ such as "dog," and the features $X'$ and $X$ represented by $P(X'|Y, \vec{Y})$ and $P(X|Y, \vec{Y})$ respectively. Assume $P$ is the LLM. In this specific category, its feature representations $X'$ and $X$ are so unique that, even though there may be some deviations, they are distinguishable. Consequently, the LLM will consistently predict them correctly, i.e., $P(Y'|Y, X, \vec{Y})$ and $P(Y'|Y, X', \vec{Y})$ are equal.

In addition, we have provided some learnability conditions for the variance of these distributions. Let $q_1$ and $q_2$ represent the data model that represents the data generation process, for instance, generating the features description given the category and the features $X$. Specifically, given that the variance of $X \sim q_1(X|Y)$ and $Y' \sim q_2(Y'|X)$ is small, this implies that the variations within the features $X$ conditioned on $Y$ and the supervision signal $Y'$ conditioned on $X$ are minimal. This further supports the assumption that $P(Y'|Y, X, \vec{Y}) = P(Y'|Y, X', \vec{Y})$ holds, as the minimal variance ensures that the instances $X$ and $X'$ within the same category $Y$ remain distinct yet consistently classifiable by the generative model.

## 4.1 MEMORY MASKING ESTIMATION THROUGH HINTS SAMPLES

In this section, we provide memory masking estimation using hint samples. We treat the LLM as a black-box model, which means that given an input $X$, it generates a predicted label $Y'$. We cannot retrieve any model parameter information from the LLMs; we only have access to the predictions for given queries. To estimate the ground truth label $Y$, we need to compute $P(Y' \mid X, \vec{Y})$ and $\sum_y P(Y' \mid Y = y, X, \vec{Y})P(Y = y \mid X, \vec{Y})$ according to Equation 2. The term $P(Y' \mid X, \vec{Y})$ can be obtained directly from the LLM's predictions. However, the term $\sum_y P(Y' \mid Y = y, X, \vec{Y})P(Y = y \mid X, \vec{Y})$ cannot be computed directly since we do not have access to the ground truth annotations for all samples. Nevertheless, this sum can be estimated if we have access to a small number of samples with known ground truth annotations (hint samples). Given Equation 2, we can write the estimation process as:

$$P(Y' = 2 \mid X, \vec{Y}) = \sum_y P(Y' = 2 \mid Y = y, X, \vec{Y})P(Y = y \mid X, \vec{Y}). \tag{4}$$

If we know for sure that $Y = 1$ for a given $X = x_1$, then $P(Y = 1 \mid X = x_1, \vec{Y}) = 1$. As a result, the above equation simplifies to:

$$P(Y' = 2 \mid X = x_1, \vec{Y}) = P(Y' = 2 \mid Y = 1, X = x_1, \vec{Y}). \tag{5}$$

Returning to the general equation:

$$P(Y' \mid X, \vec{Y}) = \sum_y P(Y' \mid Y = y, X, \vec{Y})P(Y = y \mid X, \vec{Y}), \tag{6}$$

under the condition that the example $X$ belongs to a specific class almost surely. For instance, if we know with high confidence that an example $X$ has the ground truth label $Y = y$, meaning $P(Y = y \mid X, \vec{Y}) = 1$, this indicates a hint sample where the true label is known. Therefore, as long as we can collect these samples where $P(Y = y \mid X, \vec{Y}) = 1$, we can subsequently estimate $P(Y' \mid Y = y, X, \vec{Y})$. As a result, we use hint samples to learn the dependence of the LLM's generated responses on the ground truth responses. By analyzing the relationship between the hint samples' true labels and the predicted labels from the LLMs, we can estimate the true $P(Y' \mid Y, X, \vec{Y})$. In our context, $P(Y' \mid Y, X, \vec{Y})$ can be obtained for these hint samples since $P(Y = y \mid X, \vec{Y}) = 1$. There are relevant works Yang et al. (2022), Han et al. (2018), Patrini et al. (2017) that exploit label dependence or transition matrices to obtain consistent classifiers. Nonetheless, these works are mainly conducted in white-box settings, assuming that model parameters are accessible and the estimated transition matrices are in probabilistic formats. In reality, the majority of advanced LLMs are only accessible via API calls, and there are many constraints on the format of the output. Thus, there is still a gap in providing a more feasible approach for estimating the transition matrix under the black-box LLM setting, which involves only applying prompting to retrieve informative information to improve the LLMs' responses. Moreover, limited work in prompt-based learning applies estimated transition matrix methods to help prevent LLMs from forgetting false positive predictions.

## 4.2 DISAMBIGUATION THROUGH HINTS BASED ESTIMATED MEMORISABLE MATRIX

### 4.3 MEMORISATION MASKING CONSTRUCTION

Initially, we have a full set of candidate labels $\vec{Y}$ for each query $X_i$, representing all possible classes. Our goal is to obtain the correct predictions by utilizing the deterministic mappings between the LLM's predictions and the ground truth labels established from hint samples. Let $D_{\text{hint}} = \{(X_i, Y_i)\}_{i=1}^{s}$ be a set of hint samples where $s = 4$. Each $X_i$ is an input for which we know the ground truth label $Y_i$. We use these hint samples to construct the memorization matrix $M$, which captures the mappings between the LLM's predictions $Y_i'$ and the ground truth labels $Y_i$. Additionally, we denote whole query samples as $D_{large} = (X_i)_{i=1}^{N}$ the initial queries $X = \{X_1, X_2, \ldots, X_s\}$ and the corresponding candidate set $\vec{Y}_{\text{Query}} = \{\vec{Y}_1, \vec{Y}_2, \ldots, \vec{Y}_s\}$. Assuming we know the total number of $K$ and the corresponding label for each $k - th$ category, therefore, for every $x$, there always exists universal candidate set $\vec{Y}$, therefore, $\vec{Y}_1 = \vec{Y}_2 = \cdots = \vec{Y}_s = \vec{Y}$. In the initial stage, we have only access to the full set label candidate set for each query $x$. We can denote the initial whole query label candidate set for all training samples as $\vec{Y}$ and it is a $s \times c$ matrix containing all ones and set $Y_{\text{True}} = \{Y_1, Y_2, \ldots, Y_s\}$, where $Y$ is an $s \times c$ matrix containing all ones.

$$\vec{Y}_{\text{Query}} = \begin{bmatrix} 1 & 1 & 1 & 1 \\ 1 & 1 & 1 & 1 \\ 1 & 1 & 1 & 1 \\ 1 & 1 & 1 & 1 \end{bmatrix} \quad \vec{Y}_G = \begin{bmatrix} 0 & 0 & 0 & 1 \\ 0 & 1 & 0 & 0 \\ 1 & 1 & 1 & 1 \\ 0 & 0 & 0 & 1 \end{bmatrix} \quad \vec{Y}_{\text{Updated}} = \begin{bmatrix} 1 & 0 & 0 & 1 \\ 0 & 1 & 0 & 0 \\ 1 & 0 & 0 & 1 \\ 1 & 0 & 0 & 0 \end{bmatrix} \quad \vec{Y}_{\text{Refined}} = \begin{bmatrix} 1 & 0 & 0 & 0 \\ 0 & 1 & 0 & 0 \\ 0 & 0 & 1 & 0 \\ 0 & 0 & 0 & 1 \end{bmatrix}$$

The matrix $\vec{Y}$ is filled with ones, representing that all classes are potential candidates. Our goal is to design a prompting scheme to enable LLMs to generate the correct annotation for each $q$ from the corresponding $\vec{y}$, transforming $\vec{Y}$ to $\vec{Y}_{\text{True}}$. Predictions by ChatGPT for each Query are $Y_1' = [0, 0, 0, 1]$, $Y_2' = [0, 1, 0, 0]$, $Y_3' = [0, 0, 0, 1]$, $Y_4' = [1, 0, 0, 0]$ and the corresponding Ground Truth Label vectors $Y_1 = [1, 0, 0, 0]$, $Y_2 = [0, 1, 0, 0]$, $Y_3 = [0, 0, 1, 0]$, $Y_4 = [0, 0, 0, 1]$. Estimated Memorisation Masking $M$

$$M = \begin{bmatrix} m_{Y_1'Y_1} & m_{Y_1'Y_2} & m_{Y_1'Y_3} & m_{Y_1'Y_4} \\ m_{Y_2'Y_1} & m_{Y_2'Y_2} & m_{Y_2'Y_3} & m_{Y_2'Y_4} \\ m_{Y_3'Y_1} & m_{Y_3'Y_2} & m_{Y_3'Y_3} & m_{Y_3'Y_4} \\ m_{Y_4'Y_1} & m_{Y_4'Y_2} & m_{Y_4'Y_3} & m_{Y_4'Y_4} \end{bmatrix} = \begin{bmatrix} 1 & 0 & 0 & 1 \\ 1 & 1 & 0 & 0 \\ 0 & 1 & 1 & 0 \\ 0 & 0 & 1 & 1 \end{bmatrix}$$

$\bar{Y}_{\text{Refined}} = G(X, \vec{Y}_{\text{Updated}})$. Let $\vec{Y}_{\text{Updated}}$ be an $s \times K$ matrix. If $Y_i' = 1$ (i.e., the prediction is the first class), the corresponding potential candidate set is the first row of $\vec{Y}_{\text{Updated}}$. We hope that the subsequent generation $\bar{Y}_{\text{Refined}}$ is same to the true class $Y$.

## 5 EXPERIMENTS

**Dataset:** In this paper, we evaluate our proposed Memorisable prompting method on a wide range of datasets sourced from Zhang et al. (2023). Each dataset contains both smaller and larger sizes. For a fair comparison, we only i.i.d sample $5\%$ from the large size dataset and test on the small size dataset. According to Zhang et al. (2023), each size dataset contains the same number of clusters. More information is provided in Table 2. For the CLINC, Massive, and MTOP datasets, different domains are used as labels to transform them into domain discovery. Bank77 Casanueva et al. (2020) is a banking dataset containing fine-grained intent categories classification for a single domain. CLINC(I) Larson et al. (2019) is a dataset for our intent detection from the supported intents, in this experiment, only in-domain ones are used. Massive(I) FitzGerald et al. (2022) and MTOP(I) Li et al. (2020) are both from MTEB Muennighoff et al. (2022). The "I" and "D" are abbreviations for intent and for domain, respectively.

| Metric
Dataset | ChatGPT-4o
StackExchange | ChatGPT-4o
CLINC | ChatGPT-4o
FINANCE11 | ChatGPT-4o
MOTE | ChatGPT-4o
Massive(D) |
|---|---|---|---|---|---|
| cot baseline | 44.72% ± 0.19% | 74.55% ± 2.73% | 59.41% ± 0.50% | 69.70% ± 1.07% | 63.99% ± 0.69% |
| Memorisable cot | 49.54% ± 0.31% | 75.76% ± 3.22% | 65.32% ± 1.73% | 69.25% ± 1.32% | 66.40% ± 0.38% |
| tot baseline | 41.62% ± 0.28% | 76.86% ± 0.18% | 65.62% ± 1.24% | 69.73% ± 0.42% | 68.33% ± 0.02% |
| Memorisable tot | 47.02% ± 0.33% | 77.73% ± 1.19% | 69.26% ± 1.84% | 69.00% ± 1.07% | 71.30% ± 0.26% |
| fot baseline | 42.81% ± 0.40% | 73.14% ± 2.33% | 55.87% ± 0.10% | 66.23% ± 0.81% | 67.49% ± 0.12% |
| Memorisable fot | 48.09% ± 0.35% | 74.60% ± 2.48% | 62.74% ± 0.89% | 67.60% ± 0.94% | 70.03% ± 0.19% |
| Vanilla | 49.94% ± 0.36% | 83.23% ± 1.11% | 66.61% ± 1.82% | 73.79% ± 0.61% | 72.04% ± 0.07% |
| Memorisable + Vanilla | 50.02% ± 0.25% | 82.70% ± 1.56% | 69.45% ± 2.30% | 73.72% ± 0.94% | 71.03% ± 0.07% |
| consistent baseline | 47.63% ± 0.37% | 81.85% ± 0.63% | 66.08% ± 1.38% | 74.00% ± 0.32% | 70.82% ± 0.02% |
| Memorisable + consistent | 48.19% ± 0.27% | 79.86% ± 1.53% | 65.86% ± 1.30% | 79.82% ± 0.90% | 69.81% ± 0.40% |
| Feedback baseline | 51.72% ± 0.27% | 79.34% ± 0.49% | 64.81% ± 1.33% | 71.93% ± 0.02% | 71.35% ± 0.29% |
| Memorisable + Feedback | 51.69% ± 0.24% | 80.21% ± 1.37% | 68.71% ± 2.68% | 71.06% ± 0.60% | 71.72% ± 0.38% |
| Correction | 47.55% ± 0.34% | 81.85% ± 0.63% | 65.58% ± 1.23% | 73.57% ± 0.47% | 70.84% ± 0.05% |
| Memorisable Correction | 49.71% ± 0.34% | 81.77% ± 1.28% | 68.92% ± 1.86% | 73.11% ± 0.68% | 71.26% ± 0.21% |

| Metric
Dataset | ChatGPT-3.5
StackExchange | ChatGPT-3.5
CLINC | ChatGPT-3.5
FINANCE11 | ChatGPT-3.5
MOTE | ChatGPT-3.5
Massive(D) |
|---|---|---|---|---|---|
| cot baseline | 49.59% ± 0.00% | 64.62% ± 1.63% | 19.16% ± 0.14% | 57.42% ± 0.40% | 60.54% ± 0.24% |
| Memorisable cot | 54.72% ± 0.17% | 68.09% ± 1.92% | 31.90% ± 0.30% | 64.91% ± 0.36% | 68.11% ± 0.43% |
| tot baseline | 40.50% ± 0.04% | 65.11% ± 0.09% | 57.87% ± 0.11% | 66.05% ± 0.05% | 62.04% ± 0.31% |
| Memorisable tot | 51.33% ± 0.03% | 69.53% ± 0.50% | 63.23% ± 0.44% | 73.63% ± 0.37% | 68.33% ± 0.50% |
| fot baseline | 46.04% ± 0.01% | 61.62% ± 1.16% | 36.77% ± 0.44% | 55.93% ± 0.23% | 58.23% ± 0.45% |
| Memorisable fot | 51.92% ± 0.08% | 63.53% ± 1.23% | 45.94% ± 0.60% | 63.94% ± 1.15% | 66.70% ± 0.86% |
| Vanilla | 51.96% ± 0.02% | 63.18% ± 1.13% | 62.55% ± 0.85% | 65.84% ± 0.47% | 62.22% ± 0.05% |
| Memorisable + Vanilla | 56.29% ± 0.04% | 67.84% ± 1.70% | 66.92% ± 1.10% | 74.36% ± 0.73% | 67.26% ± 0.12% |
| consistent baseline | 51.75% ± 0.06% | 68.90% ± 0.08% | 56.61% ± 0.34% | 68.26% ± 0.26% | 62.49% ± 0.19% |
| Memorisable + consistent | 53.96% ± 0.08% | 69.36% ± 0.19% | 57.42% ± 0.44% | 75.57% ± 0.77% | 64.63% ± 0.12% |
| Feedback baseline | 48.46% ± 0.00% | 71.63% ± 1.24% | 53.90% ± 2.94% | 71.88% ± 0.59% | 63.55% ± 0.02% |
| Memorisable + Feedback | 54.09% ± 0.04% | 75.29% ± 1.23% | 57.87% ± 3.37% | 68.05% ± 0.29% | 68.27% ± 0.40% |
| Correction | 51.81% ± 0.04% | 65.06% ± 0.83% | 55.94% ± 0.32% | 68.24% ± 0.09% | 62.81% ± 0.07% |
| Memorisable Correction | 56.60% ± 0.04% | 69.71% ± 1.13% | 61.93% ± 0.80% | 70.23% ± 0.62% | 68.33% ± 0.40% |

Table 1: The table shows the accuracy results for various methods evaluated using ChatGPT 3.5 and ChatGPT 4-o-mini on different datasets. For each method, both the accuracy (Acc) and standard deviation (STD) are reported. The highest accuracy achieved for each dataset ( combining our method with other prompting techniques) is in bold. Comparison of Accuracy for Self-Consistency, Few-Shot, Chain of Thought and Tree of Thought Methods

| Task | Name | #Classes | #data(small) | #data(large) |
|---|---|---|---|---|
| Intent | Bank77 | 77 | 3,080 | 10,003 |
| | CLINC(I) | 150 | 4,500 | 15,000 |
| | MTOP(I) | 102 | 4,386 | 15,638 |
| | Massive(I) | 59 | 2,974 | 11,510 |
| Topic | StackEx | 121 | 4,156 | 50,000 |

Table 2: We have used MTOP(I), MASSIVE(I), CLINC(D), Bank77 and StackEX. Overview of datasets across different tasks and domains with details on number of classes sizes and sample distribution.

## 5.1 BASELINE METHODS

For the experimental design, we conducted the following experiments: Baseline Method vs (Baseline Method + Our in ChatGPT 3.5 and ChatGPT 4o-mini). The learning objective is to show that Memorisable prompting can consistently perform well, and our method is universal and can be adapted to other prompting techniques. **Note( The estimated confusion matrix uses 5 % of total clean training samples and the estimated confusion is based on ChatGPT 3.5 only )**. For a more precise comparison, our work has applied a single prompting, unlike Cheng et al. (2023); Lin et al. (2023). We believe that using a single query minimises the uncertainty. Consistency Prompting Wang et al. (2023) aims to improve the response accuracy of LLMs by considering consistently generated answers through selecting multiple and diverse paths in a few-shot chain of thought approach. The problem with this method is its dependence on multiple sources of paths; even slight changes in one source's prediction can drastically impact the final prediction. The vanilla prompting

means we ask questions naively without additional information, only the query and full candidate set. The chain of thought method Wei et al. (2022) is a step-by-step illustration for the given query to the LLMs. Few-Shot Thought Prompting: Brown et al. (2020) uses a few relevant examples as illustrations in the prompt to aid the model in self-correction. Tree of Thought Prompting (Long, 2023; Yao et al., 2023) Yao et al. (2024) Long (2023) proposes the Tree of Thoughts (ToT) prompt, which encourages Large Language Models (LLMs) to engage in step-by-step exploration and self-correction by simulating different agents that communicate and provide critiques for each other.

## 5.2 EXPERIMENTAL RESULT

Our proposed Memorisable prompting method has proven effective and can be adapted to other prompting tasks, consistently improving the performance of other prompting techniques. The results in Table 3 show the improvements in accuracy achieved by combining our method with various prompting techniques. Our method has consistently improved across different datasets, illustrating its adaptability and effectiveness. More specifically, Vanilla Prompting-Random demonstrates drastic improvements, especially for MTOP (I), with a 14.2% increase and a significant improvement of 10.1% for StackEX (TM). The improvement is also notable for Few-Shot-Cot, showing a 14.2% We have conducted additional experiments using ChatGPT-4o-mini and ChatGPT-3.5 with two different random seeds on the datasets StackExchange, CLINC, BANK77, MOTE, and Massive(D). These experiments were performed for our method, Memorisable Prompting, as well as other baseline methods. Additionally, we have included (feedback)[1] and Self-Improve (Correction)[3] to demonstrate the effectiveness of our proposed Memorisable Prompting. The estimated Memorisable masking is acquired using a sample of hints and predictions from GPT-3.5 turbo. Therefore, the performance on ChatGPT-4o-mini is not very promising.

## 5.3 ABLATION STUDY

We have conducted our Memorisable prompting using the latest ChatGPT 4 model to illustrate the consistency of our proposed method across different LLMs. It shows that using the estimated masking based on ChatGPT 3.5 and applying it to ChatGPT 4 can still help improve the baseline Vanilla Prompting by 3.5% accuracy. The ablation study's results, which show GPT-4 as less effective than GPT-3.5, could be attributed to the transition matrix being estimated using GPT-3.5. The parameters of GPT-4 and GPT-3.5 are different, resulting in poor performance of GPT-4.

## 5.4 ABLATION STUDY APPLYING CHATGPT 4O-MINI ESTIMATED MEMORISABLE MASKING MATRIX

| Method | Accuracy (%) |
|---|---|
| Vanilla Prompting-Random t -ChatGPT 3.5 | 62.99 |
| Our Methods + Vanilla Prompting t -ChatGPT 3.5 | **67.35** |
| Chain of Thought Prompting-Random + ChatGPT 4 | 61.98 |
| Our Methods + Chain of Thought Prompting + ChatGPT 4 | **64.48** |

Table 3: Accuracy comparison of different prompting methods with ChatGPT 3.5 and ChatGPT 4 on BANK77

### 5.4.1 STUDY OF THE IMPACT OF HINT SAMPLE SIZE ON LLM ACCURACY

We have also conducted an ablation study to show that as more hint samples are used for estimating the Memorisable Masking Matrix, the accuracy of the LLM-generated annotations tends to improve.

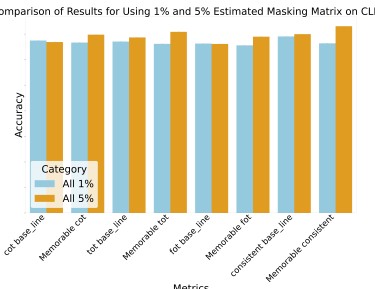

Figure 2: Accuracy for Finace77 on ChatGPT 4 in comparision with CoT and CoT+Memorisable Prompting.

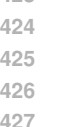

Figure 3: Comparison between using 1% and 5% of hints of total training sample estimating Memorisation Masking for datasets CLINC.

## 5.5 CONCLUSION

This paper proposes a Memorisable promoting method to prevent LLMs from forgetting positive predictions. The learning objective is to endow LLMs with the capability to memorise their past mistakes and, therefore, avoid repeating the same mistakes. We utilise small clean samples to obtain label dependence between LLMs prediction and the ground truth label. We can encode them as masking into LLMs to prevent forgetting false positive predictions. We have verified our method on a different domain dataset, showing its effectiveness across large-scale datasets.

## 6 LIMITATION

In our work, we use only a small number of clean samples (samples with ground truth annotations) to estimate Memorisable Matrix. However, for reasoning tasks, our method can be applied if the ground truth labels of the training samples are available. These ground truth labels are necessary to estimate the Memorisable mask, which can then be applied to the testing dataset. We plan to conduct additional experiments on datasets involving reasoning tasks, such as commonsense reasoning benchmarks or problem-solving datasets.

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

# A APPENDIX / SUPPLEMENTAL MATERIAL

## A.1 MEMORISABLE PROMPTING: DISAMBIGUATION PROCESS

Let $D_{\text{small}} = \{(x_i, y_i)\}_{i=1}^{s}$ be a set of small, clean samples, which is denoted as hint samples, where $s$ is the size of hint samples and $s = 10$. The $x$ is the hint query, and $y$ is the corresponding ground truth label. Additionally, we denote whole query samples as $D_{large} = (x_i)_{i=1}^{N}$ the initial queries $X = \{x_1, x_2, \ldots, x_s\}$ and the corresponding candidate set $\vec{Y}_{\text{Query}} = \{\vec{y}_1, \vec{y}_2, \ldots, \vec{y}_s\}$. Assuming we know the total number of $K$ and the corresponding label for each $k - th$ category, therefore, for every $x$, there always exists universal candidate set $\vec{y}$, therefore, $\vec{y}_1 = \vec{y}_2 = \cdots = \vec{y}_s = \vec{y}$. In the initial stage, we have only access to the full set label candidate set for each query $x$, and we can denote the initial whole query label candidate set for all training samples as $\vec{Y}$ and it is a $s \times K$ matrix containing all ones and set $Y_{\text{True}} = \{y_1, y_2, \ldots, y_s\}$, where $Y$ is an $s \times K$ matrix containing all ones.

$$
\vec{Y}_{\text{Query}} = \begin{bmatrix} 1 & 1 & 1 & 1 \\ 1 & 1 & 1 & 1 \\ 1 & 1 & 1 & 1 \\ 1 & 1 & 1 & 1 \\ 1 & 1 & 1 & 1 \\ 1 & 1 & 1 & 1 \\ 1 & 1 & 1 & 1 \\ 1 & 1 & 1 & 1 \\ 1 & 1 & 1 & 1 \\ 1 & 1 & 1 & 1 \end{bmatrix}
\quad
\vec{Y}_{G} = \begin{bmatrix} 0 & 0 & 0 & 1 \\ 0 & 1 & 0 & 0 \\ 1 & 1 & 1 & 1 \\ 0 & 0 & 0 & 1 \\ 1 & 0 & 0 & 0 \\ 0 & 0 & 1 & 0 \\ 1 & 0 & 0 & 0 \\ 0 & 1 & 0 & 0 \\ 1 & 0 & 0 & 0 \\ 0 & 1 & 0 & 0 \end{bmatrix}
\quad
\vec{Y}_{\text{Updated}} = \begin{bmatrix} 1 & 0 & 0 & 1 \\ 0 & 1 & 0 & 0 \\ 1 & 0 & 0 & 1 \\ 1 & 0 & 0 & 0 \\ 0 & 1 & 1 & 0 \\ 1 & 0 & 0 & 0 \\ 0 & 1 & 0 & 0 \\ 0 & 1 & 1 & 0 \\ 1 & 0 & 0 & 0 \\ 0 & 1 & 0 & 0 \end{bmatrix}
\quad
Y_{\text{True}} = \begin{bmatrix} 1 & 0 & 0 & 0 \\ 0 & 1 & 0 & 0 \\ 0 & 0 & 0 & 1 \\ 1 & 0 & 0 & 0 \\ 0 & 0 & 1 & 0 \\ 1 & 0 & 0 & 0 \\ 0 & 1 & 0 & 0 \\ 0 & 0 & 1 & 0 \\ 1 & 0 & 0 & 0 \\ 0 & 1 & 0 & 0 \end{bmatrix}
$$

The $\vec{Y}$ is filled with ones, representing that all classes are potential candidates. Our goal is to design a prompting scheme to enable LLM generating the correct annotation for each $q$ from the corresponding $\vec{y}$, transforming $\vec{Y}$ to $\vec{Y}_{True}$.

Predictions by ChatGPT for each Query:

$$
\begin{aligned}
Y_1' = [0,0,0,1] \quad & Y_2' = [0,1,0,0] \quad Y_3' = [0,0,0,1] \quad Y_4' = [1,0,0,0] \\
Y_5' = [0,0,1,0] \quad & Y_6' = [1,0,0,0] \quad Y_7' = [0,1,0,0] \quad Y_8' = [0,0,1,0] \\
Y_9' = [1,0,0,0] \quad & Y_{10}' = [0,1,0,0]
\end{aligned}
$$

Ground Truth Labels:

$$
\begin{aligned}
Y_1 = [1,0,0,0] \quad & Y_2 = [0,1,0,0] \quad Y_3 = [0,0,1,0] \quad Y_4 = [0,0,0,1] \\
Y_5 = [0,1,0,0] \quad & Y_6 = [1,0,0,0] \quad Y_7 = [0,0,0,1] \quad Y_8 = [0,0,1,0] \\
Y_9 = [0,1,0,0] \quad & Y_{10} = [1,0,0,0]
\end{aligned}
$$

Estimated Memorisation Masking $M$:

$$
T = \begin{bmatrix}
m_{Y_1'Y_1} & m_{Y_1'Y_2} & m_{Y_1'Y_3} & m_{Y_1'Y_4} \\
m_{Y_2'Y_1} & m_{Y_2'Y_2} & m_{Y_2'Y_3} & m_{Y_2'Y_4} \\
m_{Y_3'Y_1} & m_{Y_3'Y_2} & m_{Y_3'Y_3} & m_{Y_3'Y_4} \\
m_{Y_4'Y_1} & m_{Y_4'Y_2} & m_{Y_4'Y_3} & m_{Y_4'Y_4}
\end{bmatrix}
= \begin{bmatrix}
1 & 0 & 0 & 1 \\
1 & 1 & 0 & 0 \\
0 & 1 & 1 & 0 \\
0 & 0 & 1 & 1
\end{bmatrix}
$$

Estimated Memorisation Masking $M$ representing a potential candidate set for given an prediction label of the LLMs $Y'$ :

$$M = \begin{bmatrix} 1 & 0 & 0 & 0 \\ 0 & 1 & 0 & 0 \\ 0 & 1 & 1 & 0 \\ 1 & 0 & 0 & 1 \end{bmatrix}$$

