# OpenReview forum: "Memorisable Prompting: Preventing LLMs Forgetting False Positive Alarm"
_ICLR.cc/2025/Conference — ICLR 2025 Conference Withdrawn Submission_

### Official Review · Reviewer_m9bj · 2024-10-26

**Soundness:** 2
**Presentation:** 1
**Contribution:** 1
**Rating:** 3
**Confidence:** 3

**Summary:**

The paper introduces a method to improve the accuracy and consistency of Large Language Models (LLMs) by enabling them to "remember" past errors. It uses small annotated datasets, or hint samples, to learn dependencies between predictions and actual labels, storing this knowledge in a memory bank to help LLMs avoid repeating false positives. By applying a memory masking matrix, MP enhances prediction accuracy across domains and integrates well with various prompting techniques, making it effective for error-prone, high-stakes applications.

**Strengths:**

The strength lies in its approach to improving LLMs' reliability by addressing false positives through Memorisable Prompting (MP), which adds a layer of memory to the model's prompting strategy. This technique enhances model consistency by leveraging a memory bank that "remembers" past errors, allowing the LLM to self-correct based on learned dependencies. The approach is versatile, effectively improving accuracy across different domains and integrating well with existing prompting methods. Furthermore, the use of a memory masking matrix offers a structured way to manage and apply learned error patterns, adding a new dimension to LLM error management in a practical, adaptable manner.

**Weaknesses:**

- The article is difficult to follow due to inconsistent terminology, unclear mathematical notation, grammatical errors, structural issues, and logical gaps, which detract from its clarity and coherence. Overall, the writing does not reflect careful attention to readability and precision.

- The proposed approach appears limited in applicability, focusing primarily on classification tasks. However, this limitation is not addressed in the paper, leading to an incomplete understanding of the method's scope.

- The narrative includes redundant explanations and lacks depth, limiting the reader’s engagement with the insights behind the proposed approach.

- Baseline comparisons are insufficient; it remains unclear why traditional weak supervision methods, such as [1], combined with pre-trained language models were not included for a more comprehensive evaluation of the method.

- The proposed approach shows minimal innovation compared to prior work using transition matrices for prediction calibration, which reduces the originality and significance of the contribution.

[1] Ren, Wendi, et al. "Denoising multi-source weak supervision for neural text classification." arXiv preprint arXiv:2010.04582 (2020).

**Questions:**

Here is a revised version of your questions for the article:

- Line 120: Why not simply use $(X, Y, \mathcal{Y})$ or $(X, Y)$ here?

- Line 132: Is the concept of a "transition matrix" derived from Markov models? If so, each row in M should sum to 1. If not, consider using a different term or providing clarification to prevent potential confusion.

- If I understand correctly, Section 4 (Lines 184--241) essentially states that the smaller dataset has a similar distribution to the larger dataset, and thus the matrix M estimated from the smaller set can be applied to the larger set under certain assumptions. Why is such an in-depth discussion on conditional terms necessary if they are simply assumed valid? It seems the numbered equations and many unnumbered ones are not essential, as this is standard machine learning knowledge familiar to the intended audience.

- Line 276: Certain GPT API versions can generate token probabilities alongside tokens. Therefore, the reasoning provided here is not strong enough to justify the innovation in this paper. Additionally, relevant studies should be cited in the related works.

- Line 288: What is the difference between $D_{\text{hint}}$ and $D_{\text{small}}$ mentioned earlier? How many data points were used to compute matrix M in the experiments? It is mentioned later that $s=4$. If it accounts for 5% of the total data points, the total number of data points for each dataset should be around 80, which does not match the number in Table 2.

---

### Official Review · Reviewer_shoa · 2024-11-04

**Soundness:** 1
**Presentation:** 1
**Contribution:** 1
**Rating:** 1
**Confidence:** 5

**Summary:**

The entire paper is more of a technical report in that it improves upon the prompt strategy, which aims to make the LLMs remember the dependencies of responses and prevent making false positive answers.

**Strengths:**

The motivation of this paper is explicitly to make the LLMs remember the dependencies of responses and prevent making false positive answers.

**Weaknesses:**

The entire paper is more of a technical report in that it improves upon the prompt strategy; however, it is difficult to read the entire paper. Even after reading the whole paper, I don't even know what kind of prompt form or few-shot form the authors employ.

The motivation of this paper is explicitly to make the LLMs remember the dependencies of responses and prevent making false positive answers. However, there are many fundamental errors in both model construction and experimental setups.

In method construction:

1. The paper is very redundant to introduce the basic definition, and there is a lack of formal definition of basic memorization matrix in preliminaries.

2. The introduction of the method is not clear, I think only two points need to be introduced clearly:

2.1. How is the memorization matrix constructed between the answer and the truth to each query when the truth cannot be obtained during reasoning? How many times does each query need to be answered, and how is the sparsity of the matrix addressed when there are too many categories?

2.2. Does the LLM actually capture dependencies between responses from the matrix? How does the LLM remember and avoid false positive responses? Are there any experiments to prove it?

3. In Section 4.0, the paper introduces the learning conditions, it is very confusing how the sample features are introduced in the method, for example in Eq.3, how to find completely different features that can make the LLM  generate the same prediction? I don't think any specific distinct feature alone can do that.

4. Is it using the memorization matrix of few-shot hint samples to predict samples with no truth labels? What is the number of hint samples for each sample? **How to determine the LLM is not mimicking the memory matrix, but actually recording the response to each query.**

In related work:
The paper does not fully investigate the related work and has no logic in expression. The paper is based on LLM reasoning and constructs a strategy to improve prompt.

In the experiments:


1. The paper **is lack of experimental setup**, and prompts should be introduced in detail, otherwise it is difficult to reproduce the work.

2. There is too little experimental analysis in the paper, and the experimental Tables and Figures are not clearly introduced. Besides, the paper adopts three LLMs. Is there no difference in performance among the three LLMs? Meanwhile, the paper lacks interpretability experiments to validate the motivation.

3. References to Figures1, 2 and 3 and Tables 1 and 2 are absent from the main paper. There are also several methods that are not detailed in baselines. What is used for correction? What is fot in the Tables? **I think the paper should at least be standardized and clear.**

**There are multiple errors in the paper**, multiple quotes, punctuation, basic grammar errors, **the readability of the paper is poor**, I think the author should take the paper submission seriously.

**Questions:**

Refer to Weaknesses

---

### Official Review · Reviewer_3yDF · 2024-11-10

**Soundness:** 1
**Presentation:** 2
**Contribution:** 1
**Rating:** 1
**Confidence:** 3

**Summary:**

The main motivation behind this paper is memorizing past mistakes would make LLMs not make the same mistakes again. The authors propose Memorizable Prompting (MP), which allows LLMs to understand response dependence patterns and store them in a memory bank to prevent repeating false positive predictions. The memory bank is constructed using a small labeled set of samples.

**Strengths:**

The main idea of the paper, which states that memorizing false positive samples would help prediction performance, is interesting and can be a good direction for improving test-time techniques to improve model performance. The datasets and baselines examined in the experiments section is fairly comprehensive.

**Weaknesses:**

The justification for the Memorization masking assumption is confusing (line 225-234). Please lay out the specific constraints assumed made by this claim, and deduce the final assumption. If I understand correctly, this assumption is claiming that assuming if the model has learned the characteristics of a category well enough, then it should reliably predict the same Y' distribution for each Y regardless of the input feature X. The current justification is not clearly or rigorously written and need further editing.

Section 4 is hard to follow and not rigorously written. X and Y are sometimes referred to as variables and some other times as values/examples. For example, equation (4) does not "simplify" to equation (5) when the given assumption (ln. 258) is Y=1 for a given X=x1; this is only correct if the given assumption is X=x1 and Y=1.

Section 4.3. It is not clear how M is used in the transformation from $\vec{Y}$_Query to $\vec{Y}$_True. Additionally, "Our goal is to design a prompting scheme to enable LLMs to generate the correct annotation for each q from the corresponding $\vec{y}$" (line 303-304). What exactly is the prompting scheme? It is not discussed or included in the appendix.

Overall, there are fundamental flaws in the probabilistic framework and problem formulation. The core mathematical error lies in the proposed marginalization over Y: $P(Y'|X,\vec{Y}) = \sum_Y P(Y'|Y,X,\vec{Y})P(Y|X,\vec{Y})$. This equation is fundamentally incorrect as Y represents a ground truth label, not a random variable. The main objective to "obtain" $P(Y|X,\vec{Y})$ through $P(Y|Y',X,\vec{Y})$ shows a fundamental misunderstanding of the causal relationship in supervised learning. THe prediction Y' should not influence the probability of the true label Y. The learning objective should instead be formulated as maximizing the probability of correct classification, $\arg\max_G P(Y=y^*|X,Y)$. These issues reflect a fundamental misunderstanding of probabilistic modeling, making the proposed method mathematically unsound. While the research direction may be promising, the current formulation requires substantial revision to establish a valid theoretical foundation.

**Questions:**

1. What does it mean by justified approach through the lens of probability? (ln.76). Please be more concrete.
2. Please fix formatting errors in the related work section, the dataset section, section 5.1, etc.
3. "Given new inputs, ... refining out candidate sets and improve LLM's accuracy for subsequent generation." (ln. 178-179). Just to clarify, is this referring new inputs from D_small or D_large? Does that mean after each sample, you need to know the correctness of the produced output?
4. line 209 ChatGPT typo.
5. line 231 incomplete sentence.
6. What is P? Are you using P as a random variable, a model or a notation for probability? What do you mean "Assume P is the LLM" (ln. 232)?
7. Is X a single feature or a feature class? or a variable? It is sometimes referred to as a features and other times as features (Section 4.1), but in many equations X is referred to as a variable (e.g., X=x_1 in Eq. 5).
8. Section 4.3: What is K, what is a category? When is the concept of category introduced? What is the difference between a category and a label? Is Y a category or a label?
9. Undefined symbols: What is $K$, $c$, $\vec{Y}_G$, $q$, $\vec{y}$?
10. Is $Y_i'$ a vector or a value? It is referred to as a vector in lines 305-306 but a value 1 in line 313. Additionally, could you clarify what does it mean by "If $Y_i'=1$, the corresponding potential candidate set is the first row of $\vec{Y}_{\text{Updated}}$"? Does the value of $Y_i'$ correspond to "first row" of the matrix? Why?
11. "i.i.d. sample from the large size dataset" (ln.321)  --> do you mean uniformly sample?
12. Table 1. Bolding is mentioned in the caption but no values are bolded in the table.
13. "The problem with this method is its dependence on multiple sources of paths; even slight changes in one source’s prediction can drastically impact the final prediction." (ln. 374-375). Why is this the case? Self-consistency takes the mode of multiple reasoning paths. Furthermore, the "using a single query minimises the uncertainty" (ln. 371-372) is not well-justified.
14. All of the figures and Table 1 are not referred to in the text of the paper.

---

### Note · Authors · 2024-12-13

**Comment:**

Dear Reviewers and Area Chairs,

Thank you very much for your detailed and constructive feedback on our paper "[Memorisable Prompting: Preventing LLMs Forgetting False Positive Alarm]" (Paper ID: [I 6711]). We appreciate the time and effort you have invested in reviewing our work.

After careful consideration of your comments, we have decided to withdraw our paper at this stage.

Thank you again for your valuable feedback.

Sincerely,

The Authors

**Withdrawal Confirmation:**

I have read and agree with the venue's withdrawal policy on behalf of myself and my co-authors.